# Ecological Basis of Ecosystem Services and Management of Wetlands Dominated by Common Reed (*Phragmites australis*): European Perspective

Hana Čížková [1,*], Tomáš Kučera [2], Brigitte Poulin [3] and Jan Květ [2,4,*,†]

1   Faculty of Agriculture and Technology, University of South Bohemia, Studentská 1668,
    CZ-370 05 České Budějovice, Czech Republic
2   Faculty of Science, University of South Bohemia, Branišovská 1760,
    CZ-370 05 České Budějovice, Czech Republic
3   Tour du Valat Research Institute, Le Sambuc, 13200 Arles, France
4   Global Change Research Institute, Academy of Sciences of the Czech Republic, Bělidla 986/4a,
    CZ-603 00 Brno, Czech Republic
*   Correspondence: hcizkova@fzt.jcu.cz (H.Č.); jan.kvet@seznam.cz (J.K.)
†   Present address: Vrchlického 320, CZ-379 01 Trebon, Czech Republic.

**Abstract:** The common reed (*Phragmites australis*) is a frequent dominant species in European wetlands. Yet, its performance can vary in response to different combinations of environmental factors. This accounts for *P. australis* decline on deep-water sites, its stable performance in constructed wetlands with subsurface horizontal flow and its expansion in wet meadows. Reed stands provide habitats for nesting, feeding or roosting of vulnerable bird species. Conservation measures aim at preventing or stopping the decline of *P. australis* stands, increasing their micro-habitat heterogeneity and reducing the reed penetration into wet meadows. Service-oriented measures aim at providing suitable conditions for direct use of reed stalks for roof thatching or as a renewable energy crop or the use of the reed-dominated habitats for waterfowl hunting, cattle grazing or fishing. The compatibility between nature conservation and different socioeconomic uses can be promoted by collective agreements, agri-environmental contracts or payments for ecosystem services of the reedbeds. In situations with multiple uses, a modelling approach considering the participation of all the stakeholders concerned can be a useful tool for resolving conflicts and developing a shared vision of the respective socio-ecosystem.

**Keywords:** biodiversity; conservation measures; Europe; habitats; multiple uses; *Phragmites australis*; socioeconomic uses; wetland

## 1. Introduction

The common reed (*Phragmites australis* [Cav.] Trin. ex Steud.) is a common wetland plant species with a nearly cosmopolitan distribution, forming monodominant and productive stands under optimal conditions [1]. In its native range, local populations of *P. australis* have formed an integral part of wetland vegetation. Wetlands dominated by *P. australis* have for long provided local human communities with food (waterfowl, venison, fish), fodder and otherwise useful plant materials [2,3].

The current controversy in the perception of *P. australis* on a global scale is linked to its expansion to ecosystems with less competitive dominants and, above all, its invasion outside its native range. The invasion of genotypes of European origin in North American wetlands has stimulated research of the genetic diversity within the species and the whole genus worldwide [4,5], the ecophysiological behaviour of the invasive as compared to native genotypes (e.g., [6,7]), the ecological background of invasiveness of this species (e.g., [8,9]) and the methods of controlling its expansion ([10] and references therein). Such

studies partly overshadowed the research progress dealing with its more balanced role and management practices in its native range.

The aim of this paper is, therefore, to give an overview of various uses and ways of managing the *P. australis*-dominated wetlands in Europe, where it is native and its use has a long tradition. In the largely drained European continent, such wetlands still occupy vast areas in northern, southeastern and southwestern parts, and scattered fragments occur in the whole territory. They fulfil regulation, habitat, production, and information functions, as listed by de Groot et al. [11], and provide related ecosystem services [3,12–16]. In response to the continuing wetland drainage, support of biodiversity of wetland biota has increased in priority in the last 50 years. The management goals then reflect human preferences based on the perception of the ecosystem (Figure 1).

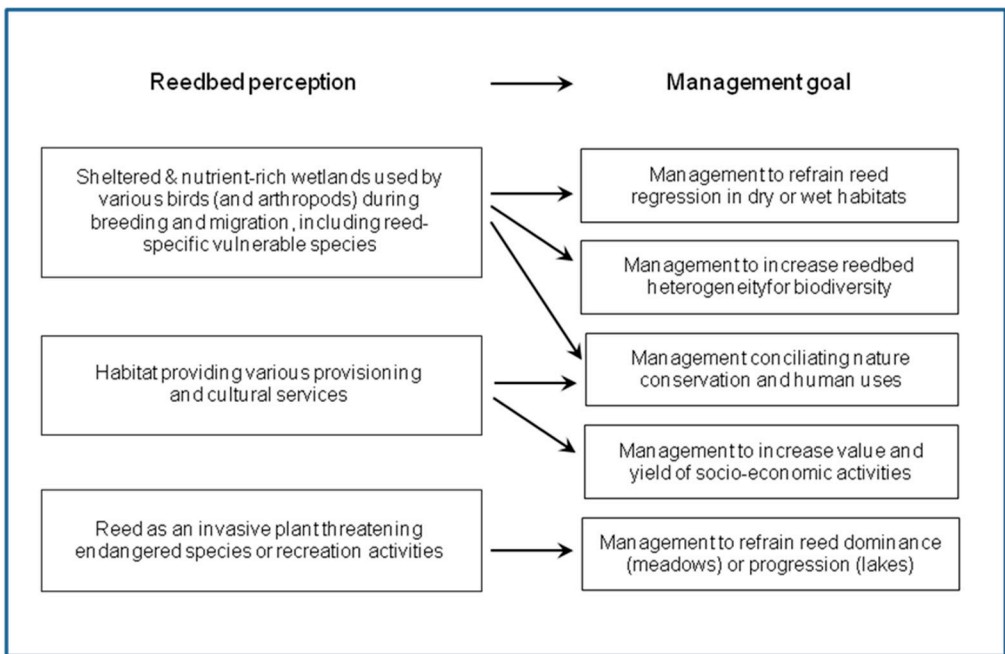

**Figure 1.** Typical management goals associated with each of the three main perceptions of reedbeds in Europe.

In the following text, we first give a brief overview of the ecological requirements and natural vegetation types with *P. australis* occurring in Europe, as background knowledge needed for their successful management. Then we focus on various uses of the *P. australis*-dominated wetlands and related management measures.

## 2. The Genetic Delineation and Ecological Niche of *P. australis* in Europe

Recent genetic studies have delimited five species of the genus *Phragmites* [5], of which *Phragmites australis* (Cav.) Trin ex Steudel is the most widely distributed. Within this species, Lambertini et al. [17] identified two genetically distinct groups of populations occurring in Europe: one inhabiting temperate Europe (European *P. australis*) and the other found in the Mediterranean region (Mediterranean *P. australis*). These two genetically delimited groups probably correspond to two respective subspecies: *P. australis* ssp. *australis* (also including the invasive populations of European origin in North America) and *P. australis* ssp. *altissimus*, proposed by Clayton [18]. Because the ecological literature scarcely distinguishes between the lower taxa of *P. australis*, only the species name (*P. australis sensu lato*) is used in this paper, except where the lower taxa are explicitly mentioned in the studies cited.

The response of *P. australis* to its habitats has been treated in detail by at least three monographs [16,19,20], two successive reviews in the series "The Biological Flora of the British Isles" [21,22] and several other review articles on the biology and ecology

of *P. australis* worldwide [23,24]. Worthy of special attention is also the conceptual article by Eller et al. [9], focused primarily on the ecological genetics of reed.

Briefly, *P. australis* is a robust perennial grass species with a nearly cosmopolitan distribution and a great capacity to acclimate to a wide range of environmental conditions regarding latitude (up to 70° north), altitude (in Europe up to 1900 m in the Alps), climate (oceanic to continental), water table (more than 2 m depth in European lakes with a great light transparency), substrate (mineral to organic), trophic conditions (oligotrophic to eutrophic), pH (2.5 to 9.8) and salinity (up to 65‰ over short periods). On the other hand, *P. australis* stands do not tolerate a sudden high rise of the water table, avoid strongly reducing organic substrates and are highly sensitive to mechanical damage of any type. Coincidence of marginal values of more factors is even more destructive.

There is also a distinct clinal variation across latitudes. European *P. australis* populations from lower latitudes tend to allocate less aboveground biomass to leaves and more to stems as compared to those from higher latitudes; they also produce fewer shoots. In the Mediterranean region *P. australis* can reach heights of up to 5 m, while in temperate Europe *P. australis* usually reaches maximum stem heights of 2–3.5 m. This relationship, however, is not linear, which is partly due to genetic differences between the temperate and Mediterranean groups of *P. australis* and is further complicated by the existence of several ploidy levels, which are not clearly related to the production and growth characteristics [9,17].

## 3. Vegetation with *Phragmites australis*

### 3.1. General Overview

*P. australis*-dominated communities represent an important long-term stage in successional seres and form important azonal wetland habitats, especially on shores of standing and slowly flowing meso- to eutrophic waters with bottom sediments and/or soils ranging between nutrient-rich and nutrient-poor ones [25]. *P. australis* is a frequent dominant or co-dominant species in communities extending along fresh and brackish running waters from their upper reaches (Figure 2A) downstream and cover large areas in river floodplains (Figure 2B–E). The largest stands of *P. australis* occur in inundated freshwater and brackish reed marshes in deltas of the main European rivers, such as the Rhine, Ebro, Rhone, Danube, Dnipro and Volga [26]. *P. australis* also forms monodominant stands in littoral zones of both natural and artificial shallow lakes (Figure 3). It is a co-dominant or dominant species of marshy fens (Figure 4A–D) and can form patches on temporarily wet hilly slopes (Figure 4E). *P. australis* is a common species in the understory of alder (Figure 2A) and willow carrs or wet pine forests [27]. It also forms an important vegetation component of wetlands significantly altered by humans and novel ecosystems ecosystems in the sense of [28], emerging in response to human activities (Figure 5). They include permanently or temporarily wet landscape elements such as constructed wetlands used for wastewater treatment (Figure 5A), drainage canals and ditches (Figure 5B), abandoned wet meadows, wet parts of spoil heaps and brownfields, and also littoral zones of artificial water bodies serving various purposes (Figure 5C,D).

Based on the phytosociological approach predominantly used in continental Europe, *P. australis*-dominated communities are included in the class *Phragmito-Magnocaricetea* Klika et Novák 1941. This class comprises vegetation types commonly occurring all over Europe and Asian Russia [29]. The class *Phragmito-Magnocaricetea* consists of 11 alliances with a total of 90 associations, out of which *P. australis* is dominant in three, constant in 34 and present in 70 ([30], Table 1). The most common is the alliance *Phragmition australis* Koch 1926, which includes associations dominated by tall helophytes, 11 of them dominated by a single tall helophyte species common in Europe. These plant stands are sometimes referred to as reedbeds *sensu lato* in ecological literature. The association *Phragmitetum australis* is the most widely spread one and often forms a mosaic with associations dominated by other common tall helophytes such as *Typha latifolia*, *T. angustifolia* or *Schoenoplectus lacustris*, and with various sedges (*Carex* spp.) near the reedbeds' landward boundaries.

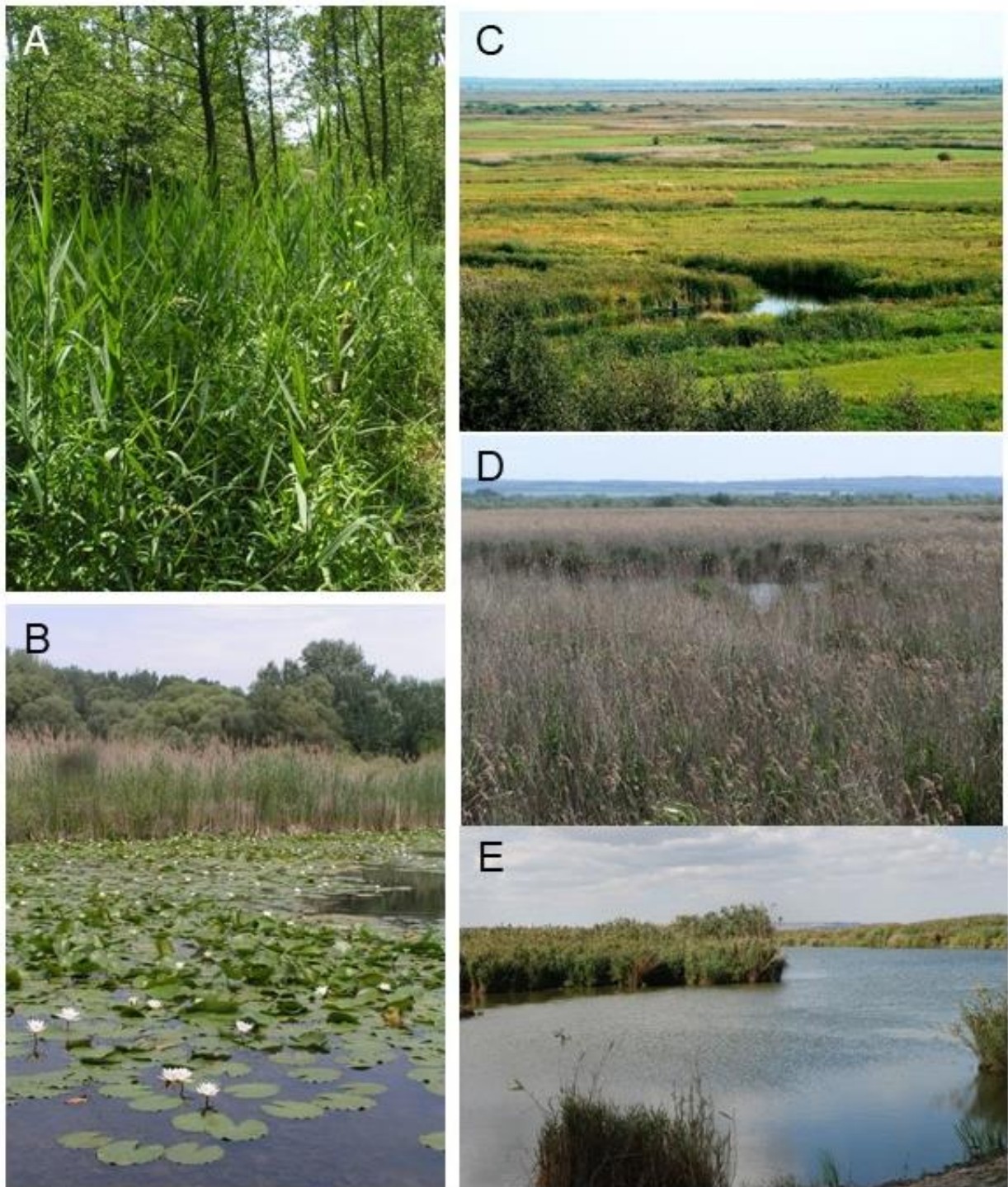

**Figure 2.** Habitats with *P. australis* along flowing waters. (**A**)—Alder carr on the upper course of the Rudava River, western Slovakia. (**B**)—vegetation zonation with *P. australis* in a eutrophic riverine habitat: the Danube, southern Slovakia. (**C**)—*P. australis* dominated non-tidal riverine wetlands: Biebrza River, eastern Poland. (**D**)—Fenéki lake, a restored *P. australis*-domianted wetland in the Kis-Balaton water protection system, Hungary. (**E**)—*P. australis* dominated tidal brackish wetlands: the Danube delta, Romania. Photographs by Hana Čížková (**A,B**), Aat Barendregt (**C,D**), Josef Rajchard (**E**).

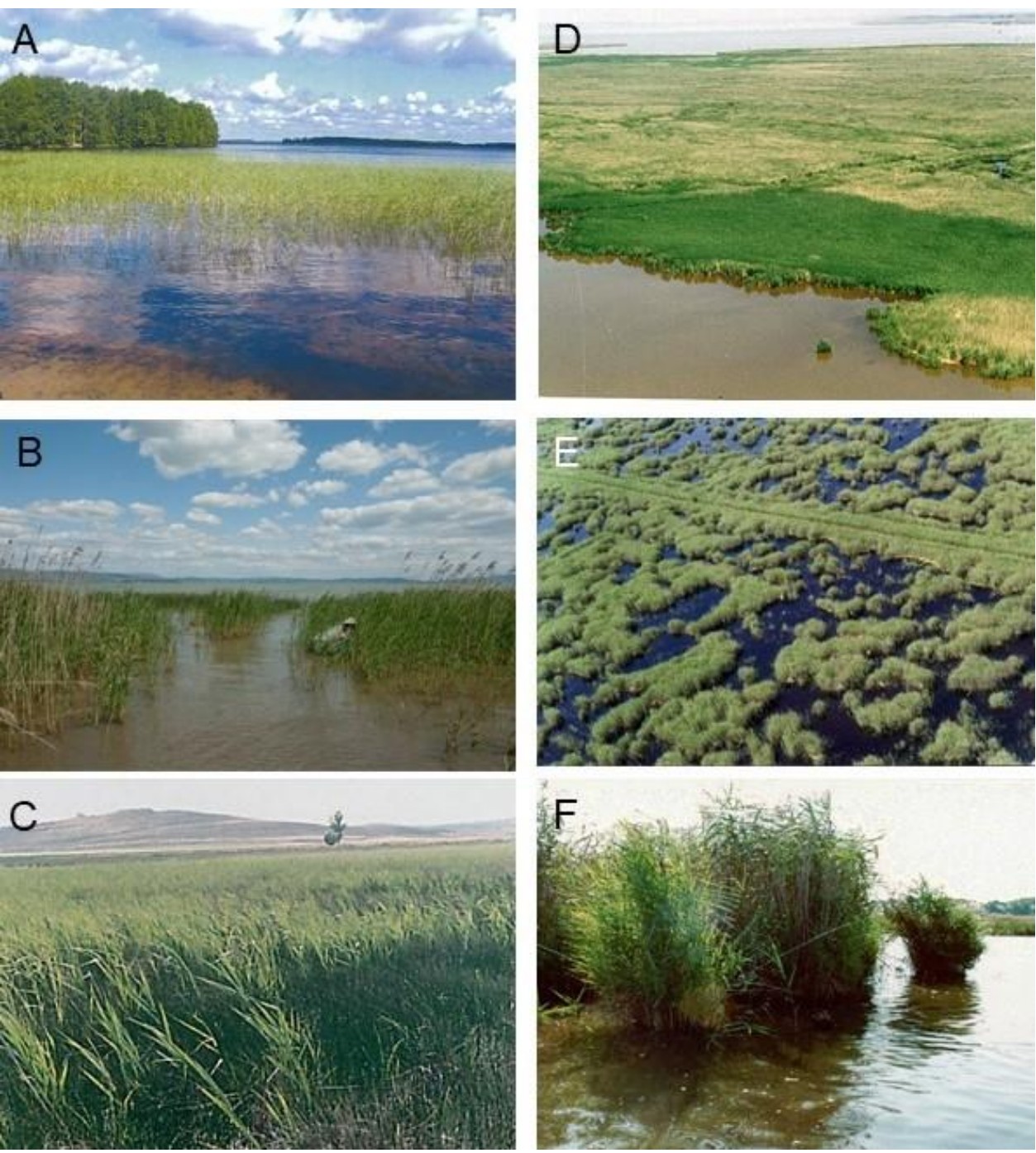

**Figure 3.** *P. australis*-dominated littoral wetlands. (**A**)—Lake Ladoga, Russia. (**B**)—declining reed stands of Lake Trasimeno, Italy. (**C**)—*P. australis*-dominated littoral zone of the saline lake Gallocanta, Spain. (**D**,**E**)—stable and declining reed stands of Lake Fertö/Neusiedlersee, Hungary. (**F**)—regenerating *P. australis* stand of Řeřabinec fishpond, Czech Republic. Photographs by Galina A. Elina (**A**), Aat Barendregt (**B**), Jiří Dušek (**C**), Mária Dinka (**D**,**E**), Hana Čížková (**F**).

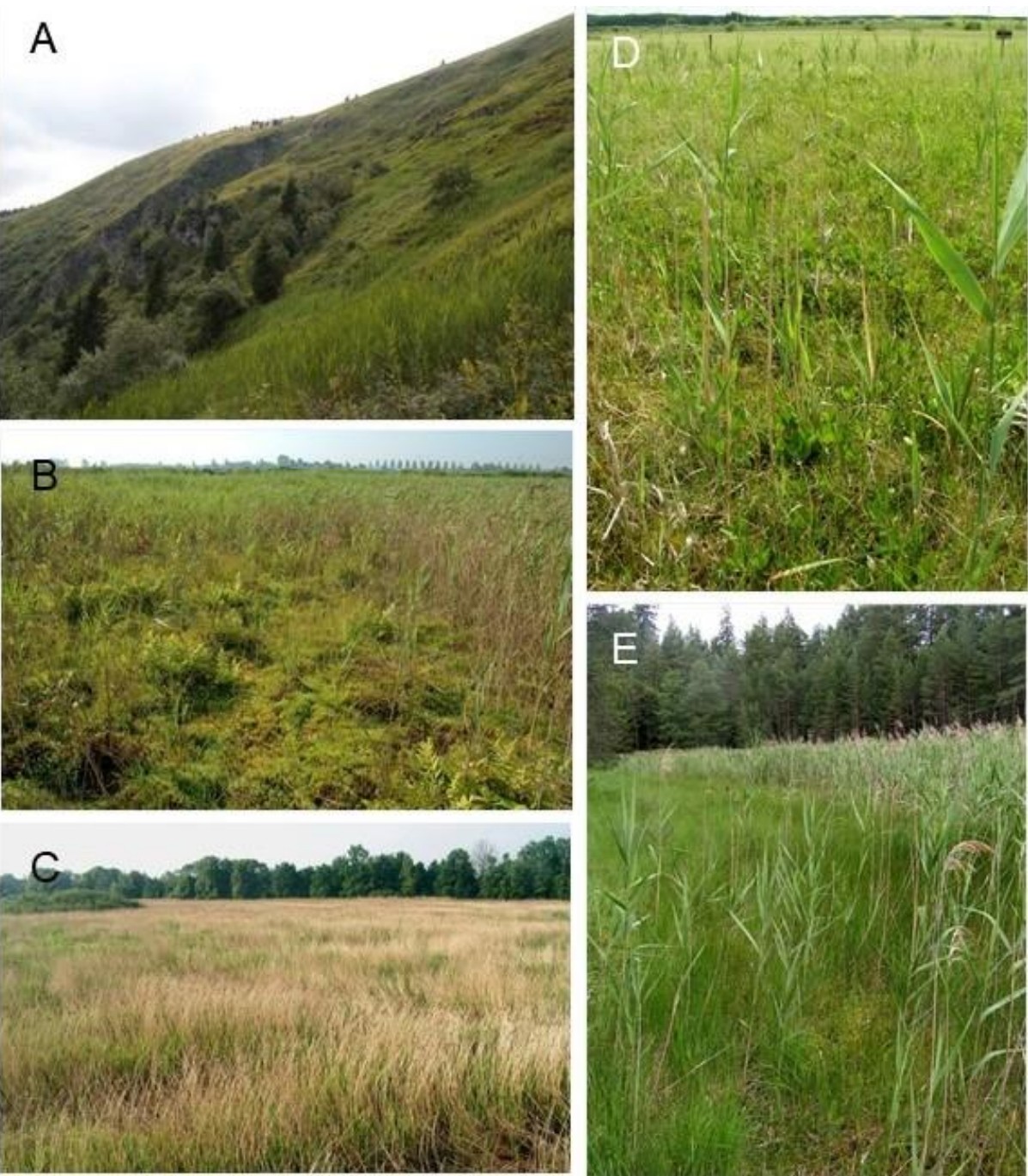

**Figure 4.** *P. australis*-dominated terrestrial habitats. (**A**)—Upper limit of occurrence of *P. australis* in the Krkonoše mountains, Czech Republic. (**B**)—*P. australis* in a non-tidal acid fen Ilperveld north of Amsterdam, the Netherlands. (**C**)—*P. australis*-dominated fen (nature reserve „U Vomáčků“), Czech Republic. (**D**)—expansion of *P. australis* in a floating fen, Rzeczin. Poland. (**E**)—Expansion of *P. australis* in a littoral sedge marsh: Staňkovský lake, Czech Republic. Photographs by Michaela Čepková (**A**), Aat Barendregt (**B**), Hana Čížková (**C**–**E**).

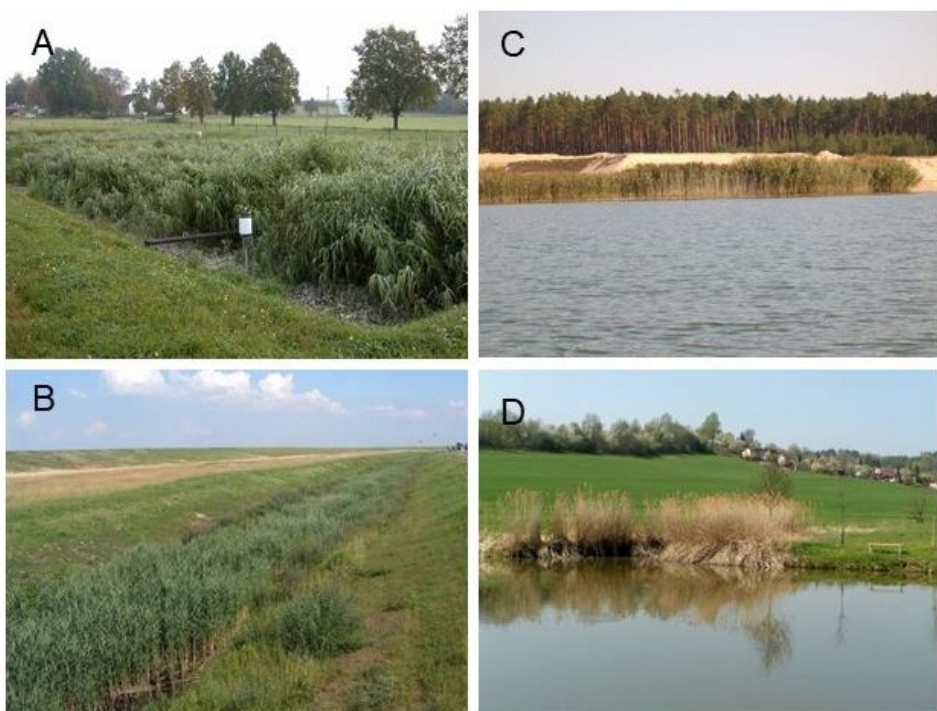

**Figure 5.** *P. australis* dominated habitats created or strongly altered by human activities: (**A**)—constructed wetland used for wastewater tratment of Slavošovice village, Czech Republic. (**B**)—drainage canal below the Gabčíkovo reservoir, Slovakia. (**C**)—*P. australis* domianted littoral of a sand-pit lake, Czech Republic. (**D**)—*P. australis* domianted littoral zone of a small village pond, Czech Republic. Photographs by Hana Čížková.

**Table 1.** Occurrence of *P. australis* in the alliances of the class *Phragmito-Magnocaricetea* based on the synpotic table published by Chytrý et al. [30].

| Sub-Class | Alliance | No. of Associations | Occurrence of *P. australis* | | | No. of Relevés |
|---|---|---|---|---|---|---|
| | | | Dominant | Constant | Present | |
| *Phragmitetalia* | *Phragmition communis* | 19 | 1 | 5 | 18 | 12,690 |
| *Bolboschoenetalia* | *Scirpion maritimi* | 7 | 1 | 7 | 7 | 1682 |
| | *Bolboschoeno maritimi-Schoenoplection tabernaemontani* | 6 | 1 | 6 | 3 | 1796 |
| *Magnocaricetalia* | *Magnocaricion elatae* | 17 | 0 | 9 | 17 | 4452 |
| | *Magnocaricion gracilis* | 6 | 0 | 3 | 6 | 5181 |
| | *Carici-Rumicion hydrolapathi* | 3 | 0 | 3 | 2 | 983 |
| *Nasturtio-Glycerietalia* | *Glycerio-Sparganion* | 9 | 0 | 0 | 0 | 3177 |
| | *Caricion broterianae* | 3 | 0 | 0 | 0 | 367 |
| *Oenanthetalia* and *Arctophiletalia* | *Eleocharito palustris-Sagittarion sagittifoliae* | 18 | 0 | 1 | 17 | 4956 |
| | *Alopecuro-Glycerion spicatae* | 1 | 0 | 0 | 0 | 30 |
| | *Arctophilion fulvae* | 1 | 0 | 0 | 0 | 19 |
| Total | 11 | 90 | 3 | 34 | 70 | 35,333 |

*3.2. Regional Survey*

In northwestern Europe, characterized by the Atlantic climate, *P. australis* grows mostly in shallowly inundated or permanently waterlogged habitats, especially in the communities of the alliance *Phragmition communis*. An overview of the vegetation with *P. australis* on the British Isles has recently been published by Rodwell [31] and Packer et al. [22]. Briefly, it is dominant in four types of wet habitats (Table 2):

1. Freshwater reedbeds usually hosting species-poor plant communities including *P. australis*, other marsh dominants such as *Typha latifolia*, *T. angustifolia*, *Schoenoplectus lacustris*, *Bolboschoenus maritimus* and tall sedges.

2. Tall-herb species-rich fens with *Cladium mariscus* and *Calamagrostis canescens* or some other species (*Juncus subnodulosus*, *Carex elata*, *C. acutifomis*, *C. appropinquata*, *C. lasiocarpa*, *C. diandra*) as co-dominants.

3. Saline brackish marshes in which more halophlous species such as *Atriplex prostrata*, *Juncus gerardii*, and *Aster tripolium* co-occur with *P. australis*.

4. A tall-herb vegetation of abandoned moist-to-wet meadows, including tall herbaceous dicotyledons such as *Eupatorium cannabinum*, *Angelica sylvestris*, *Lythrum salicaria*, *Cirsium palustre*, *Filipendula ulmaria*, and *Epilobium hirsutum*.

Additionally, *P. australis* grows sparsely in some other habitats such as salt marshes and dune slack communities on peaty mineral soils with *Salix repens*. It also frequently outcompetes sedges in fen and wet meadow vegetation in lowland regions. It occurs also in the Atlantic wet heath vegetation in the underlayer of *Hippophaee rhamnoides* scrubs on moving coastal dunes, in the understory of willow carrs, alder and willow woodlands, and birch and pine open-bog woodlands. In vegetation affected by human activities, *P. australis* occurs in tall-herb "nitrophilous" stands with *Urtica dioica*, *Cirsium arvense* and *Epilobium hirsutum* [31].

**Table 2.** Synopsis of vegetation with dominant *P. australis* in Europe: survey of habitats based on the regional vegetation monographs.

| Region/Country | Freshwater Reed Beds | Brackish Swamps | Tall-Herb Fens and Moist Meadows |
|---|---|---|---|
| N and NW Europe | | | |
| Scandinavia [32] | *Schoenoplecto-Phragmitetum* | *Bolboschoenetum maritimi* | *Magnocaricion* |
| Great Britain [22,31] | *Phragmites australis* comm. | *Halo-Scirpion* *Elymion pycnanthi* *Ammophilion arenariae* | *Phragmites australis-Peucedanum palustre* comm. *Phragmites australis-Eupatorium cannabinum* comm. |
| Netherlands [33] | *Typho-Phragmitetum* | *Phragmition* | In more communities |
| Central Europe | | | |
| Germany [34] | *Scirpo-Phragmitetum* *Phragmiti-Euphorbietum palustris* | In more communities | *Thelypterido-Phragmitetum* *Phragmiti-Caricetum lasiocarpae* |
| Poland [35] | *Phragmitetum australis* | *Phragmition* | *Thelypteridi-Phragmitetum* |
| Czech Republic [36] | *Phragmitetum australis* *Phragmition australis* | *Astero pannonici-Bolboschoenetum compacti* *Schoenoplectetum tabernaemontani* | *Thelypterido palustris-Phragmitetum australis* *Magno-Caricion elatae* *Cladietum marisci* |
| Austria [37] | *Phragmitetum vulgaris* *Phragmiti-Euphorbietum palustris* | *Bolboschoeno-Phragmitetum communis* (inland salt marshes) | *Caricion lasiocarpae* |
| SE Europe | | | |
| Hungary [38,39] | *Phragmitetum communis* *Scirpo-Phragmitetum* | – | – |
| Romania [20,40] | *Scirpo-Phragmitetum* | *Phragmition* | – |
| Croatia [41] | *Phragmition* | | *Caricetum vesicariae* *Phalaridetum arundinaceae* |

**Table 2.** *Cont.*

| Region/Country | Freshwater Reed Beds | Brackish Swamps | Tall-Herb Fens and Moist Meadows |
|---|---|---|---|
| | | E Europe | |
| Ukraine [42] | *Phragmitetum communis* | *Phragmiti-Juncetum maritimi* | *Phragmiteto-Schoenetum ferrugunei* [43] |
| Russia [44–46] (Volga [29]) | *Phragmition communis* *Calystegio-Phragmitetum* | *Puccinellio-Phragmition* *Argusio-Phragmitetum* | *Phragmiti-Magnocaricion* – |
| | | S and SW Europe | |
| France [47] | *Phragmition* (*Scirpo-Phragmitetum*) | *Phragmites communis-Juncus maritimus-Scirpus maritimus* comm. | – |
| Italy [48] | *Phragmitetum australis* | *Bolboschoenus maritimus* agg. community *Schoenoplectetum tabernaemontani* | *Magno-Caricion elatae* |
| Spain [47] | *Typho angustifoliae-Phragmitetum australis* *Scirpo lacustris-Phragmitetum* | *Scirpo compacti-Phragmitetum australis* | – |

Notes: *Phragmites communis* and *P. australis* are synonyms; we use the community name in the original form as used in the regional vegetation survey without any correction according to the Code of phytosociological nomenclature. Other synonyms: *Scirpus lacustris* = *Schoenoplectus lacustris*, *Scirpus maritimus* and *S. compactus* = *Bolboschoenus maritimus* (syn. *B. compactus*). The more detailed a regional vegetation survey is, the greater number of associations is distinguished. The negative information (–) means that either the community is not present in the region, or if present, has not been recognized and classified.

In the Netherlands, phytosociologists report *P. australis* from nearly all the habitat types described for the British Isles. In addition, they mention its occurrence in pioneer vegetation on strandlines of sand beaches and ephemeral vegetation on salt mud and sand flats [32].

In northeastern and central Europe, characterized by sub-Atlantic climate, *P. australis* grows in much the same habitats as described for northwestern Europe.Dense monodominant stands in mesotrophic to eutrophic shallow or standing water bodies are typical, alternating with other communities of tall helophytes, such as *T. angustifolia*, *T. latifolia* and *Schoenoplectus lacustris* (Table 2). Such stands occupy the transition (ecotone) between the terrestrial and aquatic zone (eulittoral to infralittoral in the sense of Hutchinson [49]). Towards open water, *P. australis* is successively replaced by diverse floating-leaved and submerged species such as *Nuphar lutea*, *Potamogeton* sp. div., *Hydrocharis morsus-ranae*, *Ceratophyllum demersum* and duckweeds (*Lemna* spp.) ([45], Figure 2B). Towards the terrestrial end of the zonation, the *P. australis*-dominated communities typically change to vegetation dominated by sedges species such as *Carex elata*, *C. acuta* or *C. riparia* ([45,50], Figure 4E). It occurs also in swamps dominated by alder (*Alnus*) and ash (*Fraxinus*), as well as in willow (*Salix* spp.) and alder carrs (stands; Figure 2A). It is a co-dominant of a variety of minerotrophic peat habitats, together with sedges such as *Carex nigra* or *C. rostrata* ([36,43,45], Figure 2C). Throughout central Europe, *P. australis* forms successional stages in abandoned meadows ([25], Figure 3E) and invades *Carex*-dominated marshes and fens in response to eutrophication ([51], Figure 4E).

In European regions with continental or Mediterranean climate (much of southern, southeastern, and eastern Europe), extensive reedbeds are associated with standing or slowly flowing fresh waters and brackish estuaries (Figure 2E), where *P. australis* can be as tall as 9 m. Floating islands dominated by *P. australis*, first described from the Danube delta [20,52], are a characteristic phenomenon in the lower reaches of large eastern European rivers [52,53]. *P. australis* is also present in inland salt marshes dominated by tall herb vegetation and on stabilized sand dunes along the sea coast (Figures 3C and 6) [54]. *P. australis* is present in all communities of the *Phragmition* alliance of the Volga River floodplain as well as in alluvial salt meadows, where it occurs together with *Argusia sibirica*, *Suaeda confusa*, *Atriplex calotheca*, *Lepidium latifolium*, *Crypsis schoenoides*, *C. aculeata*, *Bolboschoenus maritimus* agg., *Althaea officinalis*, and *Aeluropus prudens* [44].

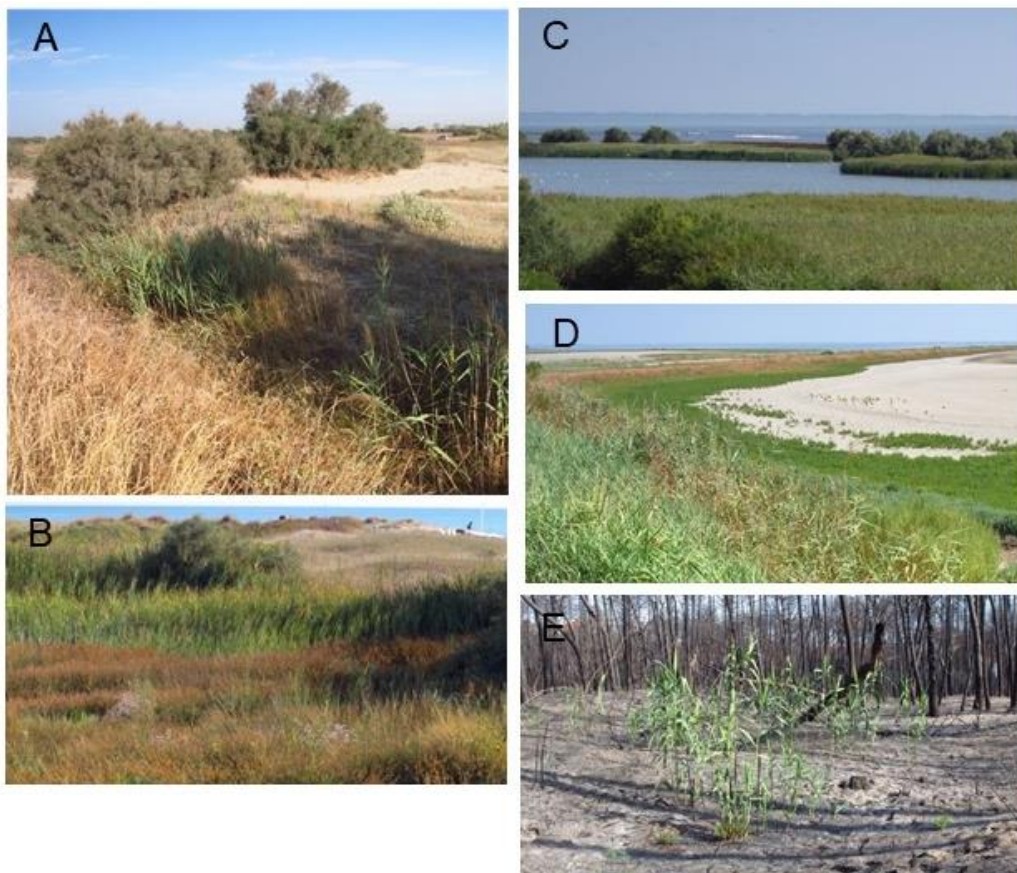

**Figure 6.** Local variability of *Phragmites australis* salt wetlands near Ravenna, Italy: (**A**,**B**)—salt marshes in back dunes with *Tamarix*, *Juncus maritimus*, *Bolboschoenus maritimus* agg. and *Limonia* [55]. (**C**,**D**)—Valle di Comacchio, brackish marshes in a coastal basin separated from the sea by a sandbank, the colony of flamingo can be seen in the middle of (**C**). (**D**)—the zonation of bank salt marsh vegetation (*Salicornia*, *Spartina* and *Sarcocornia* zones below the *Phragmites* zone. (**E**)—Lido di Dante, resprouting of *P. australis* stems five weeks after a forest fire, occurring in small local depressions with *Sallix* in *Pinus pinaster* stands in Pineta Ramazzoti (August 2012) (see also [56]). Photographs by Tomáš Kučera.

This overview indicates that *P. australis* dominates mesotrophic to eutrophic habitats subjected to long-term waterlogging or flooding, where it has its ecological optimum. From such habitats it spreads to marginal ones, suboptimal with respect to water or nutrient supply. According to the information available, its occurrence in marginal habitats is common in areas with an oceanic climate, where it is found in almost all types of wetlands, also including vegetation affected by former or current human activities (fens, abandoned wet meadows). In contrast, in areas of Europe with a continental climate, *P. australis* seems to be largely confined to habitats with sufficient water and nutrient supply.

## 4. Use and Management of *P. australis* Habitats for Biodiversity
### 4.1. P. australis Stands as Habitats of Birds and Invertebrates

Due to its vigorous growth and effective vegetative spreading, *P. australis* forms dense stands providing sheltered and nutrient-rich habitats suiting various birds and invertebrates [2,57–76]. They serve as breeding or overwintering habitats or migration stopover areas for numerous bird species including rare and endangered ones [77]. Some bird species almost exclusively use reedbeds for these purposes. They include several species of *Acrocephalus* warblers (*A. melanopogon*, *A. arundinaceus*, *A. scirpaceus*, *A. schobaneus*) and, notably, the aquatic warbler (*A. paludicola*) which is vulnerable at the global level, as well as various heron species of which many populations are depleted or still declining [77],

such as the little bittern (*Ixobrychus minutus*), the Eurasian bittern (*Botaurus stellaris*), and the purple heron (*Ardea purpurea*). Reedbeds are also extensively used as night roosts by passerines [58,78,79] and provide foraging and nesting sites to ducks and coots, the abundance of which is correlated with the reedbed area [80].

The value of *P. australis* stands as biotopes of waterfowl and other animals has been increasingly appreciated in Europe during recent decades. The restoration, or even creation, of *P. australis*-dominated wetlands has taken place mainly in western Europe, where large reedbeds have disappeared [81]. Most of the European large reed stands are now included in the inventory of Ramsar wetlands of international importance or Special Protection Areas under the European Union Bird Directive. Examples of highly valuable Ramsar sites are the Broadlands in eastern England [82], Lake Constance in Germany, Austria and Switzerland [83], Lake Neusiedlersee/Fertö in Austria and Hungary [84–86], the Lednice and Třeboň fishponds in the Czech Republic [87,88], the Rhone delta (Camargue) in France [89], and the Danube delta in Romania and the Ukraine [90]. Also, the largescale semi-natural treatment wetland, the Kis-Balaton Water Protection System in Hungary, is protected as a Ramsar wetland [91] because of its well-developed zonation of local wetland vegetation [92], supporting rich wildlife. In addition, there are numerous smaller sites protected by the legislation of individual countries.

### 4.2. Management to Stop P. australis Regression in Dry Habitats

If left unmanaged, moist areas overgrown with *P. australis* tend to change into terrestrial habitats (woodlands or grasslands depending on the regional climate) in a natural hydroseral succession process of wetland terrestrialization (landfilling). The terrestrialization of reed-dominated wetlands is primarily caused by their high net primary production. The annual production of both above- and belowground biomass of *P. australis* is usually greater than its decomposition and export [93–96]. As a result, dead biomass at different stages of decomposition accumulates on the site, and a substantial part of it is transformed into the reed peat [16].

Habitat maintenance at a reed-dominated successional stage is the basic approach to reedbed management [58]. The most common management practices are preventive and consist of reducing the biomass accumulation by removing the reed biomass by its mowing, burning or by litter removal in winter [59,94,97,98], ideally according to a short-term rotational scheme to reduce unfavourable impacts of such operations on birds and invertebrates [15]. At a more advanced successional stage, cutting or burning have only a small impact [99] and restoration through scrub grubbing and bed lowering may become necessary [100]. Stripping the topsoil followed by reed establishment through rhizome transfer, planting seedlings, and natural regeneration by raising water levels has been tested experimentally at several sites in the United Kingdom [101,102]. The management works have returned the reedbeds to an early successional stage to which Eurasian bitterns have responded rapidly [102].

### 4.3. Management to Revert the Regression of Reed in Wet Habitats

The causes of *P. australis* decline in aquatic habitats can be separated into three groups: (1) eutrophication, (2) high water levels, and (3) mechanical damage by various agents (see [103] for a review of case studies). In many instances, they operate simultaneously, and all have a joint hidden effect (Figure 7): insufficient aeration of belowground parts (roots and rhizomes), which ultimately leads to their death. Due to a lack of oxygen in the rooting substrate, an increasing amount of organic matter is decomposed by anaerobic bacteria, which is associated with the production of toxic metabolites such as organic acids, reduced forms of iron and manganese and hydrogen sulphide. These processes can form a self-perpetuating cycle, which can proceed long after the primary causes faded away.

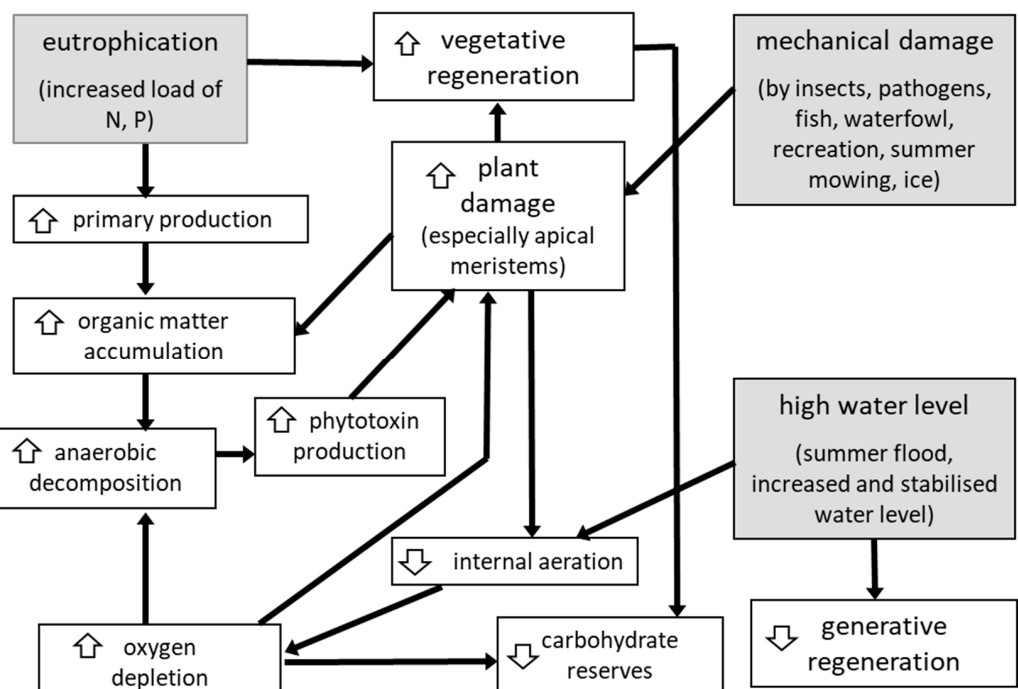

**Figure 7.** Conceptual model of factors affecting *P. australis* decline. Grey rectangles denote key environmental factors. Black arrows indicate the links between a cause and its effect. Open rectangles and open arrows indicate the processes involved and their direction, respectively.

A variety of measures have been used to reduce the nutrient load to aquatic habitats:

1. More efficient purification of wastewater discharged into the lake [104];
2. Reduction of nutrient input from neighbouring agricultural areas [105];
3. Increased nutrient stripping in the inflowing water by enhancing the mineral nutrient uptake by a dense water and bank vegetation upstream; its thereby enhanced cumulative nutrient uptake deprives the reeds growing downstream of a part of their mineral nutrient supply [106];
4. Removal of accumulated nutrient-rich mud by suction dredging [102].

The last measure can have a most rapid effect, visible within the same vegetation season, but needs to be combined with reduction of nutrient input to make the effect long-lasting.

The detrimental effects of eutrophication or of high water levels can be alleviated by winter or summer drawdown [107]. A severe summer drawdown with the water table reaching 0.5 m below ground surface during at least one month appears as the most sustainable and efficient way to reverse anaerobic conditions, especially strong in nutrient rich organic sediments. Temporary drawdown brings oxygen into the soil and thus reverses the toxicity of reduced compounds [108,109], which in turn supports the stability of the reed stands [110–113]. Experiments have shown that such a deep drying of the sediment rapidly stimulates recolonization of reeds [107]. This management is recommended at least every 5–10 years in southern France to prevent reed regression in marshes flooded permanently in order to attract waterfowl and, especially, stimulate the formation of colonies of nesting purple herons.

*P. australis* stands can also be destroyed by mechanical damage caused by human recreational activities, boat transport or mechanical effects of waves. Mechanical damage is also caused by insect infestation [22,114,115]), grazing by geese or swans, extremely dense fish stocks in fishponds, or proliferation of exotic mammals such as the muskrat (*Ondatra zibethicus*) and coypu (*Myocastor coypus*). These mammals can destroy significant amounts of reed and will severely limit its vegetative regrowth [116]. For instance, a breeding pair of muskrats can destroy nearly 1000 kg of reed per hectare to satisfy their food and shelter

demands [117]. Control programmes to limit their proliferation are often part of reedbed management [102,118].

*4.4. Management to Increase Reedbed Heterogeneity*

Ecological requirements in terms of hydrology and vegetation structure differ among reed bird species, especially during the breeding season [58,66,76,119–123]. Habitat heterogeneity is hence the most important factor influencing reed birds, next to reedbed size [59,64,124,125]. Management practices aimed at increasing the habitat heterogeneity for wildlife commonly involve:

1. Water control to provide diverse hydrological conditions over the seasons, including spring/summer flooding for nesting birds [126,127].
2. Winter reed cutting or burning according to a rotational scheme to provide reed patches of different 'ages', offering a vegetation structure that complies with the needs of all species in the long term [15,67,128,129].
3. Creation and reprofiling of gently sloping ditches and pools to provide bird foraging habitat [69,101].
4. Hydraulic works to increase habitat connectivity for migrating fish species that use reedbeds as spawning areas [130,131].

Small reed areas offer limited possibilities for spatial heterogeneity. In such situations, priorities must be set regarding which species could be favoured based on the initial state of a site and the management options available.

*4.5. Management to Stop the Spread of Reed in Wet Grasslands*

While substantial effort has been spent on protecting or restoring *P. australis* vegetation in some deep-water littoral habitats, *P. australis* is considered a nuisance because of its expansive behaviour in some originally nutrient-poor wet grasslands [13], protected because of their floristic diversity or as habitats of vulnerable birds [132]. This happens on sites subjected recently to human-induced eutrophication. The competitive success of *P. australis* under such conditions is ascribed to its ability to make better use of surplus nutrients than the sedge species can.

Common practices to reduce reed dominance in these habitats are cattle grazing at different times of year, as well as summer, autumn or winter mowing [133,134]. A 6-year field experiment carried out in Swiss fen meadows showed that *P. australis* plants retreated from the community as a result of mowing twice a year, namely in June and September [51].

Reed progression in freshwater ecosystems is best controlled by maintaining deep vertical slopes that prevent reed colonization or by mechanically damaging the reed rhizomes. The use of a cage-wheel tractor is a common practice in the Camargue, which has been successful for 10 years. Cutting of reeds several times during the growing season exhausts the rhizome reserves. Even more effective is the multiple cutting of reeds below the water surface during the growing season, which deprives the rhizomes of oxygen [135].

## 5. Use and Management for Direct Economic Benefits

*5.1. Overview of Economic Benefits*

In the past, *P. australis* was used as a resource of material for various crafts and as a technological resource. *P. australis*-dominated wetlands also served as environments providing food such as birds and fish [3,16,136]. Some of the historical uses have lasted till now, some others have been modified or abandoned. A new impetus for *P. australis* use has been given by paludiculture, i.e., the agricultural management of peaty soils, aimed at preventing carbon loss resulting from their drainage [137–139].

The use of dry reed has a long-lasting tradition for roof thatching, fabrication of mats and production of building materials [20,122,136]. Although roof thatching declined at the end of the 19th century, it has gained in popularity over the last few decades, especially in the U.K., Ireland, Denmark, Belgium, Germany, and the Netherlands. In these regions, local reed is predominantly processed by small local producers. Much of the thatching

material for western Europe, however, comes nowadays from southeastern Europe because of its higher quality and cheaper labour [140,141].

Common reed was also an important forage crop for cattle before the agricultural revolution [122] and, locally, still is. Summer harvest has become rare, but extensive grazing remains a common practice, especially in the Mediterranean area [142].

After World War II, *P. australis* was used in pulp manufacturing in some countries of the former Soviet bloc, namely the former USSR (Krotkevich 1970 in [136]), Romania [20], Bulgaria (G. Georgiev, pers. comm.), and the former German Democratic Republic (J. Köbbing, pers. comm). However, this industry was closed after the shift of these countries to the free-market economy in the 1990s, mainly owing to high harvesting costs [3]. Sustainable harvesting is also limited by a low regeneration ability of reed stands, whose terminal buds get easily damaged by the harvesting machines unless special precautions are taken [20]. Reed harvesting is now limited also by warmer winters preventing the formation of sufficiently thick ice that would support the cutting machines, or persons carrying out the reed harvest manually.

The interest in alternative energy sources has promoted the study of *P. australis* biomass yield [143–146]) and its applicability for combustion [147–151] or biomethane production [152]. An economic evaluation revealed that profitable use of harvested reed is confined to areas with relatively cheap labour and lacking long-distance energy supply or where reed is harvested as part of habitat management [153,154]).

*P. australis* cultivation also constituted the basis of the so-called biological drainage of wet areas. It was widely employed in the conversion of drained Dutch polders to agricultural land. After the polder drawdown, reed caryopses were sown (from the air) on the bare wet sediments. Within 2 to 3 years, it became completely overgrown with dense *P. australis* stands which were then left intact for several years until they were burned (as dead shoots in winter) and afterwards ploughed into the new organic-rich soil. Afterwards, rape (*Raphanus sativus*) was cultivated there, usually for two successive seasons, gradually suppressing the remaining viable reed shoots. Agricultural use of this newly gained land could start only after this stage, and the subsequent crop rotation was adjusted to eliminate almost all remaining sparsely occurring viable reed plants (e.g., [155,156].

In many European countries, (Czech Republic, France, Lithuania, Poland, Ukraine, Hungary, Serbia, etc.) fishponds were constructed for fish farming several centuries ago. Many of them, especially large ones with extensive littoral reed stands, provide habitats of great importance for the conservation of waterfowl [157–161]. Nevertheless, those without a legal conservation status are increasingly used for waterfowl hunting [157,160,162].

Sport fishing and ecotourism are also associated with the reedbeds and littoral belts as important structural elements of the landscape. This role of the reedbeds is additional to their importance as spawning areas for fish and as sites suitable for birdwatching. These provisioning services of the reedbeds have facilitated the conservation of several large reedbed areas.

### 5.2. Management for Reed Harvesting

Reed harvesting is a specific, sustainable and socially valued economic use of reedbeds. However, cutting all dry stalks in winter deprives wintering animal species of their habitat, as well as many migratory bird species of a sufficient reed cover for breeding after their return in spring, especially in continental and northern areas [63,67]. Several management options have been proposed to counteract the negative effects of reed harvesting on wildlife. A predominance of reed harvested every other year, coupled with the retention of patches harvested on a longer rotation, is considered as an effective compromise between conservation and commercial interests in the U.K. [163]. Because dry one-year stalks protect emerging next season's green shoots from late frost, biennial cutting has been shown to produce 50–75% more reed than annual cutting in the U.K. [60]. The situation, however, is different in countries such as France, where harvesting has locally remained an important commercial activity.

As two-year stalks are considered as waste material, biennial cutting requires sorting out first- from second-year stalks, therefore being no longer economically profitable. Likewise, maintaining a mosaic of reed patches of different ages with unmanaged fragments is not commercially feasible, although it is optimal for biodiversity [15,98,125]. A 5-year experiment conducted in southern France has demonstrated that optimal dry-reed density for Eurasian bitterns is obtained one year after reed cutting, especially in marshes with a homogeneous reed cover (Figure 3). Based on these results, management recommendations to reed harvesters consist of leaving 10% of uncut reed on a rotational basis (Mediterranean region) or 20% on fixed areas (northern region). It is also recommended to maintain a dry reed fringe around water bodies to preserve important bird foraging areas and reduce local damage to the rhizomes by reed-harvesting machines. Implementation of these measures has been encouraged through Natura 2000 contracts and agro-environmental schemes but could also be promoted through ecological marketing (eco-labels).

Reed harvesters need dense homogeneous stands of current-year shoots. Water management resulting in favourable conditions for reed harvest generally consists of (1) freshwater input in spring to favour reed growth, (2) summer drawdown to improve reedbed health and ground hardness (in the Mediterranean region) and prevent rhizome buds from their growing close to or above the ground surface [20], (3) low water levels in winter to increase the length of harvested stalks and facilitate access of cutting engines. Dry and leafless reed is cut before emergence of new shoots in spring and above the water (or ice) to allow dry stalks to pursue their role of rhizome oxygenation (Venturi effect).

### 5.3. Management for Waterfowl Hunting

Presence of water is essential to ducks, but permanent flooding of ponds with little water renewal often results in eutrophication and subsequent degradation of emergent and submerged macrophytes over time [163]. Periodically exposed soil is recommended to maintain appropriate conditions for sustainable management of duck populations in standing waters (J.-B. Mouronval, pers. comm.). For instance, drying of reed beds from March to September every 2–3 years will favour the dominance of annual hydrophytes and development of graminoid and amphibious plants at the marsh edge, ensuring a good seed bank for granivorous species. A short drawdown in February–March every year or at least every 3–4 years will favour the maintenance of perennial hydrophytes that are an important food source to herbivorous birds during the winter months while reducing the eutrophication rate.

Water management associated with waterfowl hunting obviously requires flooding during (and shortly before) the hunting season. However, the most common management practices involve permanent flooding or semipermanent flooding with drawdown after the hunting season (February-March). Another important aspect of the management is the creation and maintenance of large open-water areas in the vegetation to attract ducks.

### 5.4. Sustainable Grazing

Shoots of *P. australis*, especially their youngest parts, represent a favourable source of food for both domestic and wild herbivores. Wetlands provide a valuable forage crop especially in hot and dry areas such as the Mediterranean region, where the growth of terrestrial vegetation is reduced by lack of soil water from early summer.

Grazing of reedbeds by cattle is only possible when water levels are well below the soil surface. Even so, stocking rates should be less than one animal unit per hectare to be sustainable. Flooding after grazing should be avoided in order to ensure soil oxygenation necessary for rhizome recovery [164]. With one animal per hectare from June to September, the consumption of aboveground reed biomass can reach 42%, with up to 98% of biomass loss due to trampling and additional damage [165].

In view of its deleterious effect on reeds, the compatibility of grazing and nature conservation mostly consists in reed control with respect to the reed dominance and progression. Low grazing pressures on reedbeds or adjacent habitats can contribute to

their floristic diversity and provision of habitats suitable for the aquatic warbler [166] or waders [165]. The duration and periodicity of grazing (or mowing) depend on the trade-offs between the aims of vegetation control and the resulting degree of disturbance on breeding birds, sometimes translating into a rotational scheme, insuring the provision of adequate bird habitats on a long-term basis [133].

*5.5. Compatibility with Fish Farming*

The traditional fish farming in ponds takes place in 2–3-year long cycles supporting the growth of fish from fingerlings to the market size. After the end of each cycle, the fishponds are emptied for the fish harvest. Before the agricultural revolution, they were typically dried and sowed with a summer crop every 3 to 7 years to aerate the bottom sediments and thus mineralize a large proportion of organic components of the pond mud, which becomes a sink of oxygen when it is saturated with water. Relatively high amounts of mineral nutrients are exported from each fishpond during its drawdown preceding the fish harvest. Nowadays, intensification of practices aimed at increasing the fish yields include scraping of shallow littoral areas to augment the water volume for fish (at the expense of littoral vegetation), fertilization, supply of fish feed and also water oxygenation [167–170].

Compatibility between fish farming and nature conservation involves mostly the maintenance of gently sloping shores to permit the development of the littoral belt of common reed and other helophytes so that they represent at least 15% of pond area [169,170]. Intensive management practices involving the use of fertilizers, predominance of carps with less than 10% of carnivorous species and yields above 200 kg per hectare have also been shown to decrease the conservation value of the fishponds [169,171].

## 6. Restoration and Construction of *P. australis*-Dominated Wetlands

*6.1. Rewetting of Agricultural Peat Soils*

In many lowland areas in Europe, peatlands in river floodplains, as well as along the shores of lakes and seas, were drained and converted to arable land. After the soil profiles were aerated, the organic matter accumulated during previous flooding began to decompose and the soil surface began to sink. Keeping the water table low required damming of the area and continuous pumping, which is expensive and economically unfeasible in less fertile areas. Some such areas in northern Europe were therefore rewetted and then left unmaintained. The aim of these measures was first to halt peat loss [172] and then to restore spontaneously developing ecosystems accumulating peat. As the sites were usually heavily eutrophicated as a result of mineral fertiliser application during previous agricultural use, it was usually not possible to restore the original species composition, adapted to oligo- to mesotrophic conditions. The intention was therefore to create a mosaic of helophyte communities (i.e., tall sedges and reeds) and open habitats for waterfowl [173,174]. This approach has been used, for example, in northeastern Germany in the Peene River floodplain and in northwestern Hungary in the Hanság area (which was part of Lake Neusiedl until the 18th century) [175].

*6.2. Constructed Wetlands for Wastewater Treatment*

Use of constructed wetlands for wastewater treatment has gained in popularity over the last few decades [176–178]. Most constructed wetlands in Europe are planted with *P. australis*. In the 1980s and 1990s, *P. australis* was the most frequent species planted in constructed wetlands designed with continuous subsurface horizontal flow that were used to treat wastewater in small settlements and communities [179]. Much attention was also devoted to the assessment of various functions of the plants in the treatment process. To date there is much agreement that *P. australis* affects the wetland functions positively by thermally insulating the bed surface in winter, protecting it against water erosion as well as preventing clogging, and creating microhabitats for microorganisms present in the treatment bed [180,181]. In addition, *P. australis* provides a source of organic carbon for microbial processes [182,183].

Following the finding that oxygen supply to the bed by internal ventilation systems of plants is too low as to fully meet the oxygen demand for the treatment process [184–186], attention has been devoted to systems with vertical flow in which oxygen transfer to the bed is promoted by vertical percolation of the wastewater [181,187,188]. The next technological stage, i.e., hybrid systems combining the positives of both the horizontal and vertical flow [189], have retained *P. australis* as a suitable plant species.

*P. australis* is also a common plant species of surface-flow constructed wetlands, aimed mainly at nutrient removal from nonpoint sources (e.g., [190,191]). The problem of nitrogen abatement is vital especially in marine coastal areas, where nitrogen appears to be the limiting nutrient in many situations [192,193]. The main management practice associated with this use consists of cutting and removal of aboveground reed biomass. The amounts of nutrients trapped change during the growing season, with maxima attained at the peak of the aboveground biomass in the summer months [87,194].

Besides small- to medium-scale constructed wetlands, *P. australis*-dominated vegetation covers an area of about 10 km$^2$ of Fenéki Lake, forming part of one of the largest constructed wetlands of the world, the Kis-Balaton Water Protection System, Hungary. This system of a total area of about 70 km$^2$ has been constructed on the place of former natural wetlands in the mouth of the Zala River in order to trap nutrients and suspended solids carried by its waters before they are discharged to Lake Balaton [195].

## 7. Multiple Uses

Preference for particular uses of *P. australis* stands leads to conflicts of interest among groups of various stakeholders. Problems occur mainly in harmonising the management of reed for biodiversity on the one hand and its uses for direct economic benefits on the other hand. The timing and amplitude of water-level fluctuations represent the most important complex abiotic factor. Water requirements of many breeding birds are compatible with hydrological conditions that favour reed growth in spring and support the overall stability of the plant stands. On the other hand, *P. australis* stands can retreat as a result of permanent flooding required to attract ducks for hunting or stabilized high water tables in fishponds aimed at maximising fish production. High stocks of cyprinid fish also compete with ducks for food such as zooplankton or benthos.

Many of the conflicts can, however, be prevented or overcome with management actions considering multiple benefits. Implementation of a collective agreement regarding water management rules can be necessary to favour diversity of uses and avoid ecosystem degradation. The plastic morphology of reeds, as well as the rapid yet reversible responses of reedbed structure to environmental conditions, makes it an ideal system for implementing evidence-based, adaptive co-management approaches by their users.

In situations of multiple uses with potential negative impact on ecosystem health, a companion modelling approach involving scientists and stakeholders can be useful to solve conflicts and build a shared vision of the socio-ecosystem [196,197]. The simple ecological functioning of reedbeds makes this ecosystem particularly suited for modelling [123]. An agent-based model called REEDSIM was developed in the Camargue [198] for testing long-term effects of various management schemes, climatic scenarios and market contexts on the health, biodiversity and economic yield of reedbeds (Figure 8). It comprises three sub-modules: (a) a topographical and hydrological module that defines the structural properties of a virtual wetland flooded by seasonal water levels, (b) an ecological module that sets reedbed and bird population dynamics, and (c) a decision module specific to each kind of activity, defined through semidirective interviews with each type of users (farmers, reed harvesters, hunters, and naturalists). A simplified version of the model has further been developed into a role-playing game (RPG), called BUTORSTAR, which simulates the impacts of reedbed management resulting from decisions made by the farmers, reed harvesters, hunters, and naturalists [199]. This RPG is based on an archetypal wetland made of a virtual landscape. Four different water regimes are proposed, each one adapted to a particular wetland use. Land-use and water management decisions are made by

the players at both estate and management-unit levels. These decisions are entered into the model each year as the results of the negotiation process between the players. This RPG creates a continuum of learning that crosses the traditional boundaries between disciplines and allows the players to conduct multipurpose experiments that contribute to their comprehensive understanding of the socio-ecosystem. Typically, a hunter is asked to play the role of a reed harvester and so forth, facilitating dialogue among users in situations of conflict and providing a transdisciplinary knowledge-based tool to support collective thinking and decision processes.

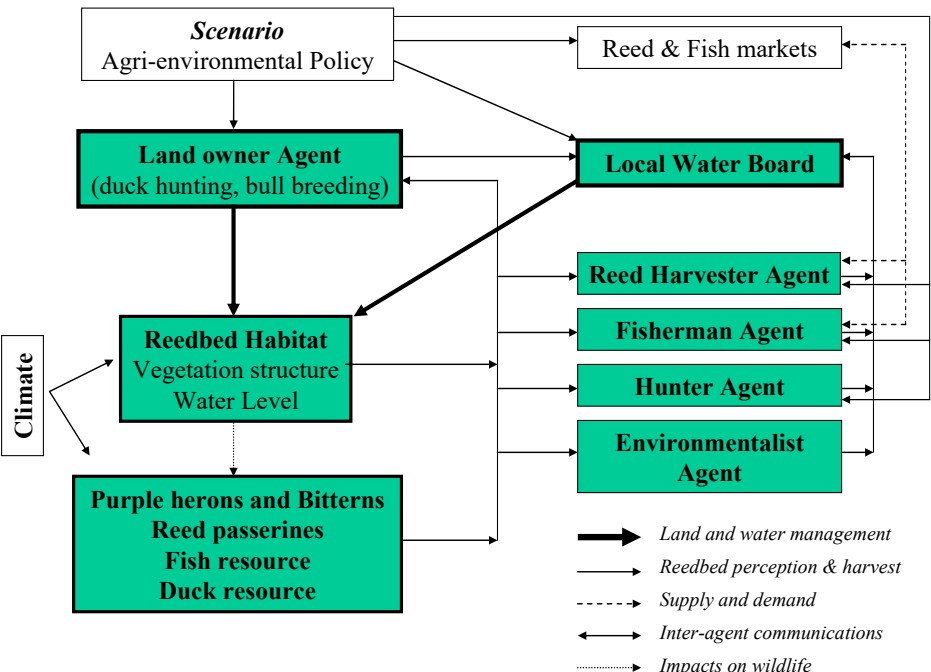

**Figure 8.** Conceptual frame of the REEDSIM agent-based simulation, (adapted with permission from Mathevet et al. [124]). This model comprises three sub-modules: the physical environment, bird population dynamics and socioeconomic decisions of stakeholders.

## 8. Future Prospects of *P. australis* in Europe

The present intensive land-use and search for adaptation measures to climate change represent new drivers of ecological development of European landscapes. If incautiously applied, they may inflict negative effects on all types of wetlands [200]. A holistic approach needs to be developed in order to counteract or, at least, minimise them.

The information reviewed in this paper clearly documents the diversity of *P. australis* habitats and human uses. This knowledge may help us predict possible changes in its status in Europe in connection with the ongoing climate change. Čížková et al. [200] have considered the likely changes to wetland biotopes. The following impacts may specifically concern *P. australis* biotopes: (1) In coastal areas, sea level rise might result in a reduction of the area of *P. australis*-dominated wetlands in estuaries of large rivers. (2) In continental areas of southeastern Europe, littoral wetlands dominated by *P. australis* may be negatively affected by anticipated water shortages. (3) In central and western Europe, the anticipated increase in the frequency and duration of flooding are likely to become a continuous threat to *P. australis* stands in lakes. (4) In northern Europe, the predicted increase in temperature might favour the expansion of *P. australis* in two ways: directly by stimulating *P. australis* growth and indirectly by increasing nutrient availability as a result of accelerated decomposition of soil organic matter. These mechanisms may be important especially in littorals of oligotrophic lakes and in wet grasslands.

As an opportunistic species of a highly competitive potential, *P. australis* will continue to occupy wet unmanaged biotopes in agricultural landscapes and occur in wet succession seres on abandoned land such as spoil heaps.

Lefebvre et al. [201] simulated the future evolution of water balance, wetland condition and water volumes necessary to maintain *P. australis* habitats at mid- and late- 21st century at 135 localities in Mediterranean Europe under two scenarios assuming a stabilization (RCP 4.5) or increase (RCP 8.5) of greenhouse gases emissions. The simulations performed under current conditions show that wetland habitats would remain in good condition at 97% of localities. However, by 2050 this proportion would decrease to 87% and 66% under the RCP 4.5 and RCP 8.5 scenarios, and even further to 78% and 36% by 2100. The simulations suggest that wetlands could persist with up to a 400 mm decrease of annual precipitation. Such resilience to climate change was attributed to the semipermanent character of wetlands (lower evaporation on dry ground) and their capacity to act as water reservoirs (higher precipitation expected in some countries during winter). The countries at highest risk of wetland degradation and loss were Portugal and Spain. Degradation of *P. australis* stands due to climate change will negatively affect their biodiversity and the services they provide as animal refuges and primary resources for industry and tourism. Preservation of their catchment areas and proactive management to reduce nonclimate stressors is urgently needed to preserve these wetlands.

As follows from previous sections, human preferences in landscape management may be equally important as environmental determinants for the further fate of *P. australis*-dominated wetlands. As pointed out by Čížková et al. [200], this holds for the future condition of European wetlands in general. Focusing on *P. australis*-dominated wetlands, the role of the species as a habitat former is particularly important in wetlands of international importance [202] and in constructed wetlands. The knowledge of ecophysiological mechanisms underlying *P. australis* performance forms a useful theoretical background for effective management of such *P. australis* wetlands. The use of *P. australis* as potential raw material and alternative energy resource appears to benefit from association of the uses with biotope care (e.g., [203]).

## 9. Conclusions

1.  This review of knowledge on European *P. australis* populations indicates that it is a plastic and versatile species, forming part of varied plant communities all over Europe.
2.  The analysis of the ecophysiological response to multiple stressors is used as a tool for understanding the population dynamics of *P. australis* in the main habitat types in Europe. Its decline at deep-water sites, stable performance in constructed wetlands with subsurface horizontal flow and expansion in wet grasslands are given as examples.
3.  Of various human uses, the role of *P. australis* as a habitat former has gained an increasing value. Vulnerable birds are major drivers of reedbed management, especially in northwestern Europe, where large reedbeds have deteriorated or disappeared, which was followed by intensive habitat management ('gardening'), restoration and creation. Traditional socioeconomic uses are being abandoned, intensified or replaced by more lucrative activities (e.g., waterfowl hunting). Uses of common reed as energy crop and renewable eco-material for green buildings are limited but promising.
4.  Each of the uses should be based on management practices that include both natural and human-driven processes. Nevertheless, the long-term maintenance or intensification of the economic uses often leads to practices that are not sustainable and get into conflict with nature conservation. Harmonisation of multiple uses with the help of innovative approaches (modelling) can assure a more sustainable future of *P. australis* wetlands.

Generally, *P. australis* will continue to be an important wetland species both in the ecological and social contexts in Europe, owing to its importance in both natural and human-altered vegetation, as well as its other ecosystem and economic values.

**Author Contributions:** Conceptualization, H.Č. and J.K.; vegetation, T.K.; conservation, B.P.; ecosystem services, H.Č. and B.P.; review and editing, J.K. All authors have read and agreed to the published version of the manuscript.

**Funding:** This research received no external funding.

**Institutional Review Board Statement:** Not applicable.

**Informed Consent Statement:** Not applicable.

**Data Availability Statement:** Data sharing not applicable.

**Acknowledgments:** We thank Dennis Whigham for his comments on the manuscript and Aat Barendregt, Jiří Dušek, and Josef Rajchard for providing photographs for Figures 2–5.

**Conflicts of Interest:** The authors declare no conflict of interest.

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
