# Peer review of "Ecological Basis of Ecosystem Services and Management of Wetlands Dominated by Common Reed (Phragmites australis): European Perspective"

_diversity, doi:10.3390/d15050629_

Round 1
Reviewer 1 Report
This paper provides an in-depth review of ecological basis informing ecosystem services of the common reed Phragmites australis from a European perspective. Overall, the paper is well written and organized. Given that it’s a review paper, it provides sufficient consideration of the published literature for the topic.
Prior to publication, I have a few points of consideration for the authors. The primary point is related to the overall conceptual model presented in Figure 6. The ‘key’ environmental factors presented affecting P. australis fail to mention any pressures from biotic sources. In the US, an emerging issue is large die-backs due to a non-native scale insect Nipponaclerda biwakoensis (e.g., Cronin et al. 2020). Although, this may not yet be a prevalent issues in the EU, I wonder if the authors fail to anticipate future biotic pressures and factors arising from invasive species and climate related changes to other unanticipated biotic pressures? Additionally, I think the authors could be a bit more specific in the terminology used for ‘phytotoxin production’. Are the authors referring to production of sulfides and acetic acid from aerobic conditions? Or is ‘phytotoxin’ meant to mean allelopathic compounds?
Cronin, J. T., Johnston, J., & Diaz, R. (2020). Multiple potential stressors and dieback of Phragmites australis in the Mississippi River Delta, USA: Implications for restoration. Wetlands, 40, 2247-2261.
Specific comments:
Line 223: Delete duplicate figure capture insert in Figure 5
Line 304: Parenthesis after (Jacoby 1970)
Line 345: “wise” – suggest an alternative word. Perhaps ‘effective’?
Line 633: The authors reference a role-playing game ‘BUTROSTAR’ from 2007. Is this still relevant to present day? Given the rapid advancements in computational modeling I would be surprised if this now 16 year old model is still a relevant topic. Are there more recent examples of models that can be given instead?
Line 736: I think the use of toxins used here is meant to describe sulfides and acetic acid? But this can be confusing because it can also be used for allelopathic compounds. Need to clarify for the reader.
Author Response
The response is provided in the attached document.

Reviewer 2 Report
Review report on MS entitled: Ecological basis of ecosystem services of marsh vegetation dominated by common reed (Phragmites australis). European:perspective; Authors: Hana Čížková , Tomáš Kučera, Brigitte Poulin and Jan Květ; MS submitted to: Diversity, MDPI
General comments
The paper is actually a geobotanical review without exemplified information, for instance, about plant density, climate condition, pedology, bird (flocks, nesting) density, fish spawning or shoals, etc. for comparable perspective. The paper is a literature survey limited to European conditions but include significant international issues which are not incorporated into the text., The major considerations of the paper are focused on beneficial aspects of Phragmites australis (PA) whilst major concerns are recently aimed at PA as nuisance. Moreover, a remarkable list of references is presented but other significant literature is missing, such as papers or books, such as, (among others) by W.J. Mitch, Hoffmann, Bakker, Gorham, and Kadlec. Out of 180 citations only 17 (9.5%) are later than 2010 and several PA issues are repeatedly presented in different chapters whilst focusing on assembled compilations regarding certain aspect is recommended.
Conclusively: The paper is recommended to be rejected in its present form and encouraged for resubmission after revision.
Specific Comments
The botanical nomenclature for Phragmites australis is:
Phragmites australis australis (sub-species is included).
Line 33: Global distribution
Lines 75-76: indication of Altitude and Latitude location is recommended
Line 101: change with by accompany is recommended
Line 141: change flowing water to running water is recommended
Line 150: the term “littoral wetland” is incorrect change to shallow is recommended.
The term “saline P. australis” is unclear, precise indication is recommended.
Figures in pages 4-7 require indication for the outstanding of each photo.
The text in pages 4-8 is of geobotanical descriptive and specific features of climate, soil type, hydrology are missing.
Line 194-204 is needless.
Lines 207-217 Justify a chapter solely on distribution.
Figure 5: Some data about salinity and density is recommended.
Lines 249-258: Exemplified data is recommended
Line 263: Characterization of Oceanic climate is recommended. What about oceanic salty aerosols?
Line 276: replace numerous by number
Line 313: insert objective or goal
Line 318: replace dead by degraded or decomposed
Line 335: Possibility: replace halt by stop, reduce, or decline.
Chapter 4.3. It is recommended to focus the entire paper about the topic: PA is a nuisance or biodiversity factor.
Line 350: What is the impact of nutrient input from agricultural vicinity on PA density.
Lines 358-380: wetting and anoxia enhance reduced condition and production of toxicity (sulfide) whilst oxygenation reduce toxicity. Clarification is recommended. A known case in the Hula Valley where Typha domingensis was demolished as a result of Phosphorus availability decline whilst toxication impact of sulfide on PA require high concentration of sulfide and long tern chronic impact.
Line 372: Myocastor coypus is exotic , not native.
Figure 6: Nice model scheme but without time frames and quantification the ecological significance value is not high..
Lines 431-455: It was significantly important in the past. What are the actual consequences and how it is correlates with biodiversity?
Line 465-475: The relevance to biodiversity is not clear enough.
Lines 476-482: PA is nuisance vegetation in fishponds as well as in lake public beaches which interfere by blocking free passage to the water front, and shelter for predator animals or even venomous snakes. And if inundated produce decomposition matters which enhance anoxia and toxic matters production which deport spawner fishs or young fingerling shoals.
Lines 503-516: How PA enhance biodiversity suppose to be the focus but how? sheltering, suitable for nesting, or food availability? Chicks treatment?
Line 527-536: The grass as food is indicated but what is the fate of further developed biomass?
Chapters 8.2. and 9 are the very relevance and making them as the central topic of the whole paper is recommended.
Line 689: The insert of sea level rise impact require information about salinity impact of PA growth.
Line 693: The issue of dryness impact on PA growth require consideration.
Lines 694-700: If this is positive or negative effect should be considered.

Author Response
The response is included in the attached document.

Round 2
Reviewer 2 Report
The revised version is recommended to be published